# Local-Global Spatial-Temporal Graph Convolutional Network for Traffic Flow Forecasting

**Xinlu Zong [1,\*], Zhen Chen [1], Fan Yu [1] and Siwei Wei [2]**

1   School of Computer Science, Hubei University of Technology, Wuhan 430068, China;
    cz1965128059@163.com (Z.C.); yuf472850@gmail.com (F.Y.)
2   School of Computer and Artificial Intelligence, Wuhan University of Technology, Wuhan 430070, China;
    waosfengw@whut.edu.cn
\*   Correspondence: zongxinlu@hbut.edu.cn

**Abstract:** Traffic forecasting's key challenge is to extract dynamic spatial-temporal features within intricate traffic systems. This paper introduces a novel framework for traffic prediction, named Local-Global Spatial-Temporal Graph Convolutional Network (LGSTGCN). The framework consists of three core components. Firstly, a graph attention residual network layer is proposed to capture global spatial dependencies by evaluating traffic mode correlations between different nodes. The context information added in the residual connection can improve the generalization ability of the model. Secondly, a T-GCN module, combining a Graph Convolution Network (GCN) with a Gated Recurrent Unit (GRU), is introduced to capture real-time local spatial-temporal dependencies. Finally, a transformer layer is designed to extract long-term temporal dependence and to identify the sequence characteristics of traffic data through positional encoding. Experiments conducted on four real traffic datasets validate the forecasting performance of the LGSTGCN model. The results demonstrate that LGSTGCN can achieve good performance and be applicable to traffic forecasting tasks.

**Keywords:** traffic forecasting; spatial-temporal features; graph attention residual network; transformer

## 1. Introduction

The increase in the number of vehicles and changes in travel patterns have imposed great pressure on urban road capacity. Intelligent Transportation Systems (ITSs) [1] present a solution to enhance transportation operational efficiency while preserving environmental resources. An integral component of ITSs is traffic forecasting, which utilizes historical data to predict future traffic flow [2]. Accurate traffic prediction not only serves as a decision-making foundation for travel planning [3] but also contributes to enhancing the efficiency of city road traffic [4].

In early days, traffic flow prediction relied on methods based on statistics [5] and machine learning [6,7]. However, their predictive performances were suboptimal due to the intricate nonlinearity of traffic data and spatial-temporal dependencies. Deep learning techniques [8–10] have emerged as effective solutions to address these challenges, showcasing significant success in diverse domains, such as target detection [11] and machine translation [12]. Presently, deep learning approaches are progressively displacing conventional traffic flow forecasting methods [13,14].

Traffic forecasting is primarily influenced by the interplay of traffic flow across time and space. Temporally, historical flow exerts a significant impact on current traffic patterns. Meanwhile, in spatial terms, the interactions among road nodes contribute to fluctuations in traffic flow. Various spatial-temporal prediction models [15,16] have emerged to extract traffic features by considering the inherent dependency in both temporal and spatial embeddings. Effectively capturing the spatial-temporal correlations during the evolution of traffic patterns is a critical factor for obtaining accurate prediction results.

Recently, the utilization of Graph Neural Networks (GNNs) has gained prominence for capturing spatial features within traffic systems, particularly in the interaction among road sensors, owing to its exceptional performance on graph structures. Additionally, the extraction of traffic spatial-temporal dependencies [17,18] has been enhanced by integrating Convolutional Neural Networks (CNNs) [19,20], Recurrent Neural Networks (RNNs) [21,22], and an attention mechanism [23,24]. Notable examples include the Diffusion Convolutional Recurrent Neural Network (DCRNN) [25] and Temporal Graph Convolutional Network (T-GCN) [26], which leverage graph convolutional techniques and the Gated Recurrent Unit (GRU) [27] to extract spatial-temporal dependencies. Although these methods have been proven effective in traffic prediction, three significant challenges remain to be addressed.

Firstly, traffic patterns between roads in the transportation network are interrelated, and two regions far away from each other in the city will also have related traffic patterns [28,29]. For example, during peak commuting hours, people often travel between residential and commercial areas. These two areas are usually not adjacent to each other, but traffic changes between the two areas are closely related. However, most traffic forecasting approaches only focus on the neighborhood information, ignoring the global cross-regional spatial dependence, and fail to completely extract spatial features. Thus, it is necessary to enhance global spatial relationship modeling.

Secondly, when an emergent event (e.g., a car accident) occurs at a road node, the traffic patterns on adjacent roads may change rapidly in a short period of time. This indicates that the neighborhood information shows stronger real-time dependency compared with the distant regions. Moreover, traffic conditions at two locations that are far apart should not affect each other within a short period of time. Thus, it is essential to extract real-time neighborhood information for the global region. However, a lot of traffic prediction approaches only consider the static spatial relationships between roads and ignore the dynamic influence between neighbors [30]. As shown in Figure 1a, the accident at the central road node makes the adjacent road nodes 1 and 2 impassable, but the vehicles at nodes 3 and 4 still pass normally. This indicates that traffic flow should be dynamically influenced by the spatial relationships between roads. It is not enough to capture spatial features only based on the static distance between roads.

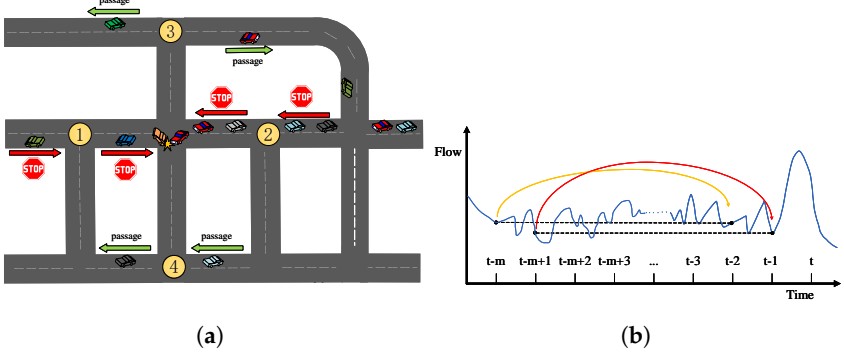

|  (**a**)  |  (**b**)  |

**Figure 1.** Impact of changes in traffic and flow. (**a**) Different traffic modes on adjacent roads in case of accident. (**b**) Long-term correlation of traffic flow on roads.

Thirdly, GRU and Long Short-Term Memory (LSTM) [31] have been widely adopted in various spatial-temporal sequence modeling. However, in those models, a fixed long recurrent structure leads to extremely long input and output components. The long-term temporal dependence cannot be extracted effectively due to gradient vanishing [21,30]. As shown in Figure 1b, the flow values at the timestamp t-m and t-2, pointed out by the yellow curve, and the timestamps t-m+1 and t-1, pointed out by the red curve, have similar distributions. Although they are far apart, there may be hidden correlations in them. Therefore, the global temporal dependence of traffic data is also important in traffic forecasting.

This paper introduces a novel traffic forecasting framework based on Local-Global Spatial-Temporal Graph Convolution Network (LGSTGCN). The LGSTGCN framework encompasses three key components: a Graph Attention Residual Network layer, a T-GCN module, and a Transformer layer. These components are designed to capture the spatial dependence of global regions, local spatial-temporal characteristics, and global temporal dependence, respectively.

The primary contributions of this article can be summarized as follows:

- A graph attention residual network layer is proposed for capturing global spatial dependencies, and richer information about the spatial semantics of different channels can be learned through residual connections.
- A module based on T-GCN is added for extracting real-time spatial-temporal dependence. The graph convolution of the module can learn the dynamic influence of neighboring roads in the current time step.
- A transformer layer is presented for extracting the global temporal dependence of the traffic flow data. The positional encoding scheme used in the layer enables the included self-attention mechanism to identify the position characteristics of the traffic sequence.
- Experiments of different baseline methods, different hyperparameter settings, and ablation study of model modules are conducted. The experimental results show the accuracy and validation of the proposed model.

The remaining sections of this article are structured as follows: In Section 2, we conduct a review of the existing literature pertaining to traffic prediction. Section 3 provides a definition of the traffic forecasting problem. Section 4 details the presentation of the LGSTGCN model. In Section 5, the experimental comparison between the baseline approach and the LGSTGCN model is analyzed. In Section 6, the conclusion is drawn and future research is discussed.

## 2. Related Work

### 2.1. Traditional Approaches

At the outset of traffic forecasting, it was initially approached as a regression statistics issue. Early classical statistical methods like the Autoregressive Integrated Moving Average (ARIMA) [5] model and Historical Average (HA) [32] model were utilized. However, these statistically based approaches, which rely on the assumption of stationarity, fail to adequately capture the highly nonlinear nature of traffic data.

Consequently, machine learning approaches such as the K-Nearest Neighbors (KNN) [6] algorithm and Support Vector Regression (SVR) [7] algorithm have been employed for traffic forecasting. These approaches proved to be more efficient; however, they relied heavily on manually designed features. Nonetheless, traditional methods generally fail to capture the complex spatial-temporal characteristics of traffic systems.

### 2.2. Deep Learning Approaches

Deep learning methods, particularly those leveraging techniques such as Recurrent Neural Networks (RNNs) and its variants like GRU, have shown excellent performance in traffic prediction. This is attributed to their adaptive learning capabilities concerning the temporal and spatial features of traffic data. To address the intricate temporal aspects of traffic patterns, RNNs and its variants like GRU are favored for their superior sequence information processing. For example, Ma et al. [33] introduced an LSTM-based prediction method, effectively capturing traffic sequence correlations. Fu et al. [31] utilized both LSTM and GRU to forecast traffic flow. In addition to RNN-based approaches, certain CNN methods have found application in traffic forecasting. Zhang et al. [13] employed LSTM and CNN to capture spatial-temporal characteristics for traffic prediction. Another approach by Zhang et al. [34] transformed the traffic network into an image grid, extracting spatial correlation through residual convolution cells. It is worth noting, however, that

CNNs are more suitable for two-dimensional grid spaces, whereas real road networks exhibit irregular topologies.

GNNs have garnered increasing attention in the context of traffic prediction. Zhao et al. [26] introduced a T-GCN model, combining GRU and GCN to extract both temporal and spatial features of traffic data. Li et al. [25] proposed a DCRNN model, which incorporated diffusion on a directed graph to simulate traffic flow. Wang et al. [35] utilized bidirectional graph message passing to capture fine-grained positional spatial interactions, employing cyclic aggregation for real-time fusion of spatial-temporal embeddings. Cui et al. [36] defined graph convolutional operators based on traffic network topology, integrating LSTM for spatial-temporal traffic prediction. Liu et al. [37] constructed a physical graph directly based on realistic road topology and built a similarity graph and correlation graph using virtual topology. All the complementary graphs are merged into the graph convolution gated recurrent unit for spatial-temporal representation learning. However, these GNN-based methods usually focus on local spatial information and ignore the global spatial dependency while extracting spatial features, which leads to missing the hidden cross-region spatial correlation features.

To address this limitation, certain studies have shifted their focus towards enhancing the representation of spatial features from local to global perspectives. For instance, Zhang et al. [29] devised a multi-scale network that integrated a graph attention network with a graph diffusion mechanism based on convolution, effectively preserving both local and global dependencies of spatial features. Zhao et al. [38] introduced a model that used adaptive correlation and spatial attention to capture local spatial dependencies and global spatial dependencies, respectively. For temporal dependencies, bidirectional gated recurrent layers and temporal attention mechanisms were utilized. Another approach by Zhang et al. [39] involved capturing spatial features from local and global perspectives using attention graph neural networks and convolutional networks. However, these methods overlooked the contextual features of global spatial dependencies, limiting the model's generalization ability.

### 2.3. Attention Mechanism

The attention mechanism, recognized for its ability to assess the significance of traffic flow data and adjust information distribution, has proven effective in aggregating data information across various research domains, including person re-identification [40] and action recognition [41]. Recently, attention mechanisms have gained widespread use in traffic prediction. For instance, Bai et al. [42] introduced the A3T-GCN model, combining GCN and GRU to capture dynamic spatial-temporal correlations. The model incorporated an attention mechanism to capture long-term temporal dependence. Ye et al. [43] developed a meta graph transformer structure for traffic forecasting by embedding meta-learning into three self-attention mechanisms. Additionally, Xu et al. [44] extracted directed spatial dependency using a self-attention mechanism and used a transformer framework to capture long-term bidirectional temporal dependencies. However, relying solely on a single attention mechanism neglects the sequential features of traffic flow. Moreover, a model exclusively based on the transformer architecture may struggle to effectively capture local information, thereby limiting the model's overall learning ability.

Based on the analysis and discussions of the current research, the primary challenges in traffic forecasting involve effectively modeling global spatial dependence and capturing local-to-global temporal dependence while preserving the sequential position information of traffic data. Aiming at these challenges, this paper introduces a new traffic prediction framework based on the Local-Global Spatial-Temporal Graph Convolutional Network (LGSTGCN). In the temporal dimension, a combination of GRU and transformer architectures is employed to extract local-to-global temporal dependence. In the spatial dimension, a graph attention residual network layer is specifically designed to capture global spatial dependence, incorporating context information. Additionally, graph convolution is utilized to further extract dynamic local spatial dependence. This comprehensive framework aims

to tackle the identified challenges in traffic forecasting by integrating both global and local spatial-temporal dependencies.

## 3. Problem Definition

Define the real traffic road topology as graph $G = (V, \delta, A)$, where $V$ is the collection of road nodes and $|V| = N$, $\delta$ is the set of edges corresponding to the connections between nodes. $A \in \mathbb{R}^{N \times N}$ is the adjacency matrix of the graph. Any elements $a_{ij}$ in A denote the connectivity between nodes $v_i$ and $v_j$ ($a_{ij} = 1$ means $v_i$ and $v_j$ are connected, $a_{ij} = 0$ means $v_i$ and $v_j$ are not connected). The traffic flow of the nodes in the graph $G$ is represented as a feature matrix $X \in \mathbb{R}^{N \times F}$, where $F$ denotes the length of the traffic flow sequence on the road.

Traffic forecasting aims to forecast future traffic sequences $(X_{t+1}, X_{t+2}, \ldots, X_{t+T})$ by using the road topology graph $G$ and the historical traffic flow sequences $(X_{t-m}, X_{t-m+1}, \ldots, X_t)$, where $M$ is the length of the historical traffic flow sequence and $T$ denotes the length of the future sequence to be predicted.

The mapping function $f$ from the historical series to the predicted series can be formulated as Equation (1):

$$f((X_{t-m-1}, X_{t-m}, \ldots, X_{t-1}), G) \rightarrow (X_t, X_{t+1}, \ldots, X_{t+T}) \tag{1}$$

## 4. Methodology

A traffic forecasting model based on the Local-Global Spatial-Temporal Graph Convolutional Network (LGSTGCN) is presented to capture the spatial-temporal dependence of traffic flow data from local to global. The general architecture of the LGSTGCN model is shown in Figure 2. The LGSTGCN model consists of the following three parts.

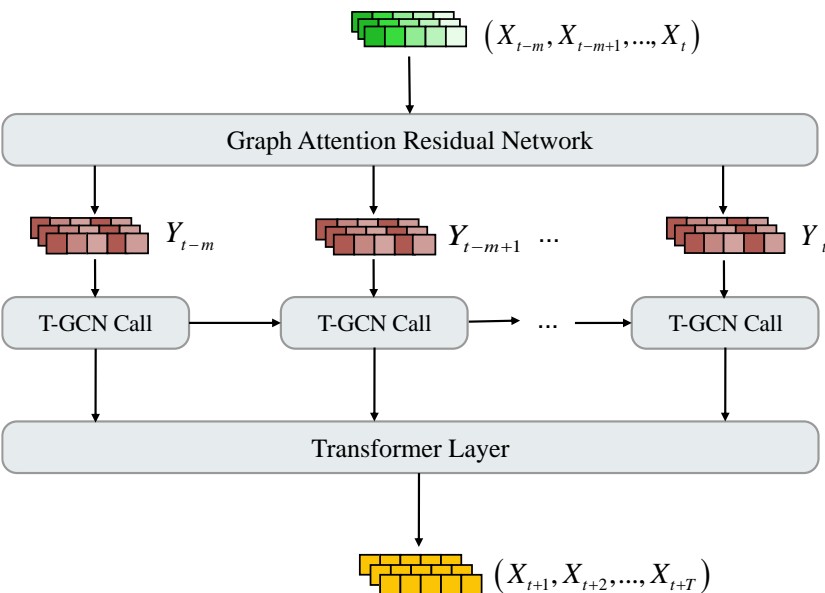

**Figure 2.** The architecture of LGSTGCN.

*Global spatial dependency modeling.* The graph attention residual network layer is designed to capture global spatial dependencies by calculating feature correlations between different nodes. Furthermore, the inclusion of spatial context information through residual connections is implemented to enhance the generalization capability of the LGST-GCN model.

*Real-time local spatial-temporal dependency modeling.* In the graph attention residual network layer, because the features of neighboring nodes can be updated according to their correlations with the central node, the graph convolution pays more attention to those

neighborhood nodes with stronger correlations when aggregating neighborhood information. In addition, combining with GRU, the real-time local spatial-temporal dependence in the current time step can be extracted.

*Global temporal dependency modeling.* The transformer layer is designed to capture the global temporal dependence of a traffic sequence. Different attention levels are designated to different positions in the traffic sequence in parallel through positional encoding.

### 4.1. Global Spatial Dependency Modeling

A graph attention residual network layer is introduced to extract the global spatial dependence of nodes' cross-regions in a traffic system, as illustrated in Figure 3. The feature matrix $X \in \mathbb{R}^{N \times F}$ is the input of the layer.

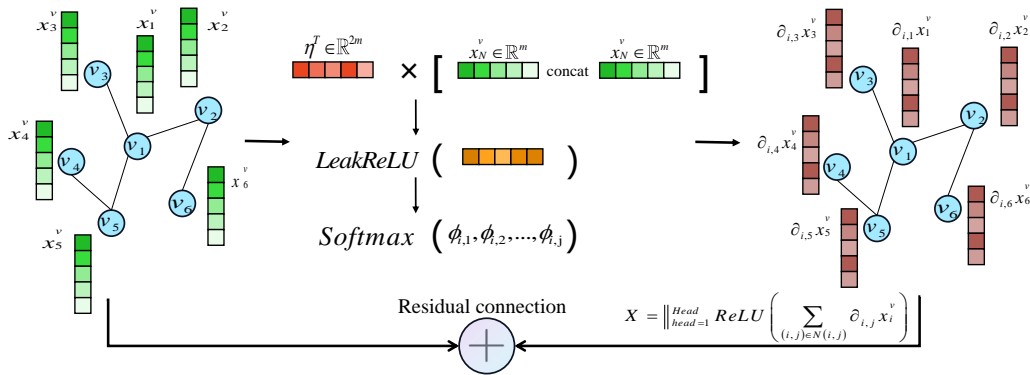

**Figure 3.** The graph attention residual network layer capturing global spatial dependence.

To obtain sufficient spatial feature representation, the feature vector $x_N^v \in \mathbb{R}^m$ of each node is mapped to a higher-level feature space $\widehat{x}_N^v$ by a linear transformation, as shown in Equation (2):

$$\widehat{x}_N^v = x_N^v W^v \tag{2}$$

where $W^v \in \mathbb{R}^{m \times m}$ denotes the linear transformation weight matrix.

The graph attention mechanism assesses the pairwise correlation between every two nodes in the traffic topology graph [45]. Define $\phi_{i,j}$ as the correlation coefficient between nodes $v_i$ and $v_j$. $\phi_{i,j}$ can be expressed as Equation (3):

$$\phi_{i,j} = LeakyReLU\left(\eta^T\left[\widehat{x}_i^v \parallel \widehat{x}_j^v\right]\right) \tag{3}$$

where $\parallel$ is the feature vector concatenation, $\eta^T \in \mathbb{R}^{2m}$ is the weight vector, and *LeakyReLU* is the activation function.

Define $\partial_{i,j}$ as the attention weight between nodes $v_i$ and $v_j$, and it can be denoted as Equation (4):

$$\partial_{i,j} = softmax(\phi_{i,j}) = \frac{exp(\phi_{i,j})}{\sum\limits_{(i,j) \in \mathrm{N}(i,j)} exp(\phi_{i,j})} \tag{4}$$

where *softmax* is the activation function. Equation (4) means that the attention weight of two nodes is associated with the correlation coefficient between them.

The attention is embedded in a multi-headed way to enrich the representation of spatial features of the model. The node feature matrix $\widehat{X}$ containing the global spatial dependency is defined as Equation (5):

$$\widehat{X} = \parallel_{head=1}^{Head} ReLU\left(\sum\limits_{(i,j) \in \mathrm{N}(i,j)} \partial_{i,j} \widehat{x}_i^v\right) \tag{5}$$

where *Head* is the number of attention heads and $\parallel$ represents the concatenation of the feature matrices of the different headspaces. *ReLU* denotes the activation function. In Equation (5), the feature vector of node $v_i$ is first multiplied with the attention weight $\partial_{i,j}$ in each subspace, and then the feature fitting is performed by using the *ReLU* activation function. Finally, the feature matrix $\widehat{X}$ that contains the global spatial dependency is obtained by embedding the feature representations of multiple subspaces jointly.

In order to enhance the generalization ability of the spatial features of the model, the residual connection is used to add spatial context information to the model. Define $Y$ as the feature matrix containing global spatial context information, as shown in Equation (6):

$$Y = \widehat{X} + X, Y \in \mathbb{R}^{N \times F} \tag{6}$$

### 4.2. Real-Time Local Spatial-Temporal Dependency Modeling

The framework of T-GCN [26] is shown in Figure 4 The T-GCN module is designed to extract real-time spatial-temporal dependence of local traffic. The GCN is embedded into a linear representation of the input and hidden state of the GRU to capture local spatial-temporal dependence in the current time step. Before graph convolution, the neighborhood nodes with greater correlation are updated by the graph attention residual network layer, which allows the graph convolution to learn neighborhood feature information dynamically instead of static features.

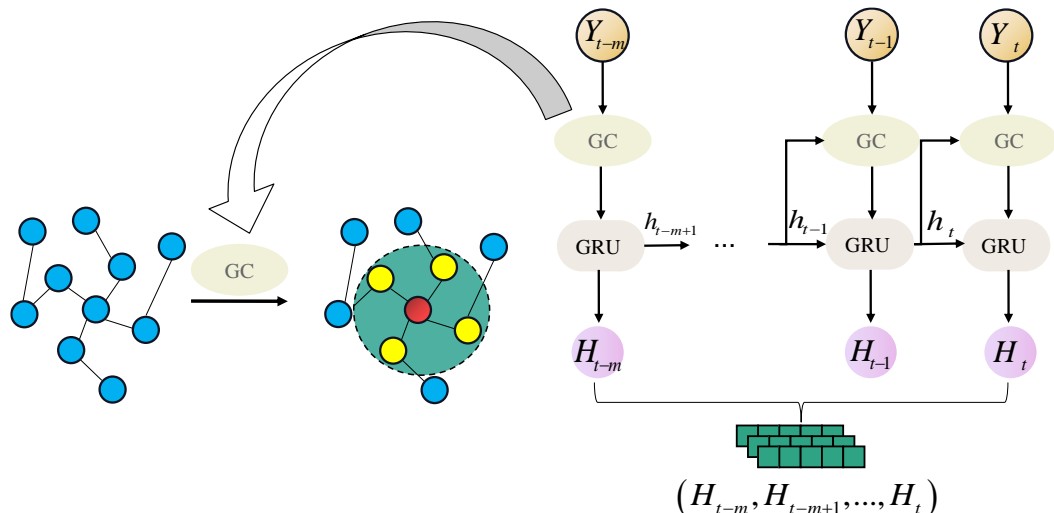

**Figure 4.** The framework of the T-GCN module.

### 4.2.1. Local Spatial Dependency

The first-order neighborhood features of aggregated nodes in the GCN layer are used to capture local spatial dependency. $GC(\bullet)$ is the graph convolution operation, as shown in Equation (7):

$$GC(X) = \sigma\left(A_{lap} X W_{gc}\right) \tag{7}$$

where $W_{gc}$ is the learnable matrix and $\sigma$ is the sigmoid activation function. The characteristic normalization of the Laplace matrix $A_{lap}$ is shown in Equation (8):

$$A_{lap} = D^{-\frac{1}{2}}(A + I_N)D^{-\frac{1}{2}} \tag{8}$$

where $I_N$ is the identity matrix and $D$ is the degree matrix of $A + I_N$.

### 4.2.2. Local Temporal Dependency

For capturing the real-time local temporal dependence, the sequence information is extracted in short time steps using the GRU. The GRU captures the temporal information

by keeping the hidden state through the gating mechanism. The linear transformation of the GRU is replaced by graph convolution. Define $u_t$, $r_t$, and $c_t$ as the reset gate, update gate, and candidate hidden state of the $l$th time step, respectively, which are calculated by Equations (9)–(11):

$$u_t = \sigma(GC(Y_t, H_{t-1}) + b_u) \tag{9}$$

$$r_t = \sigma(GC(Y_t, H_{t-1}) + b_r) \tag{10}$$

$$c_t = \tanh(GC(Y_t, r_t * H_{t-1}) + b_c) \tag{11}$$

where $b_u, b_r, b_c$ are the biases and tanh is the activation function. The hidden state $H_{t-1}$ of the previous time step $t-1$ and the input $Y_t$ are connected as the input of the graph convolution operation.

Denote $h_t$ as the hidden state of the GRU at the current time step $t$, and it is computed by Equation (12):

$$h_t = u_t * h_{t-1} + (1 - u_t) * c_t \tag{12}$$

The ultimate output of this module is a spatial-temporal feature vector $(H_{t-m}, H_{t-m+1}, \ldots, H_t)$ containing nodes.

### 4.3. Global Temporal Dependency Modeling

The GRU can capture local temporal information sequentially based on the recurrent structure. However, the GRU cannot capture long-term dependency well due to its long path between input and output components [46]. Actually, traffic temporal dependence may be correlated not only sequentially in a short period of time [30] but also over a long time interval. Therefore, a transformer layer is proposed to capture global temporal dependency. The transformer layer focuses on all time positions based on the attention approach. This global attention approach has strong modeling capabilities for long sequences, thereby effectively learning long-term temporal dependencies. In addition, the multi-head attention mechanism enables the model to focus on several different subspaces and learn different feature information. The output $(H_{t-m}, H_{t-m+1}, \ldots, H_t)$ of the T-GCN is denoted as the spatial-temporal feature matrix $H \in \mathbb{R}^{N \times F}$, which is taken as the input of the transformer layer. As illustrated in Figure 5, the transformer layer contains a multi-head self-attention sublayer, two normalization sublayers, a feed-forward neural sublayer, and a prediction sublayer.

#### 4.3.1. Positional Encoding

Parallelization of attention may lead to the neglect of the relative position of sequence information. A thermal encoding time position information is embedded into the input.

Define $\eta_{po(i)}$ as the positional encoding matrix of the $H \in \mathbb{R}^{N \times F}$ for each position $i$, as shown in Equation (13):

$$\eta_{po(i)} = \begin{cases} \sin\left(po/10,000^{i/d_{model}}\right), & \text{if } i \text{ is even,} \\ \cos\left(po/10,000^{(i-1)/d_{model}}\right), & \text{if } i \text{ is odd.} \end{cases} \tag{13}$$

where sin and cos are the trigonometric functions, $po$ is the node location, $d_{model}$ is the feature size of the transformer layer, and $i$ is the position of the node feature vector element. Define $H^p$ as the spatial-temporal feature matrix containing the thermal code, as illustrated in Equation (14):

$$H^p = H + \eta_{po(i)} \tag{14}$$

$H^p$ contains the position information of the traffic flow sequence, which makes the attention no longer treat the position of each sequence equally.

### 4.3.2. Multi-Head Self-Attention Mechanism

The scaled dot-product attention is adopted in the multi-head self-attention layer. In order to improve the model learning ability, the spatial-temporal feature matrix $H^p$ containing positional encoding is mapped into three new feature spaces, which are denoted as queries ($Q$), keys ($K$), and values ($V$), respectively. They are illustrated in Equation (15):

$$Q = H^p W^q, K = H^p W^k, V = H^p W^v \tag{15}$$

where $W^q \in \mathbb{R}^{d_{model} \times d_k}$, $W^k \in \mathbb{R}^{d_{model} \times d_k}$, and $W^v \in \mathbb{R}^{d_{model} \times d_k}$ are the matrices to be learned. The self-attention score $Attention(.)$ is defined as Equation (16):

$$Attention(Q, K, V) = Softmax\left(\frac{QK^T}{\sqrt{d_k}}\right)V \tag{16}$$

where $d_k = d_{\text{mod } el} / Head$ is the training gradient stability factor.

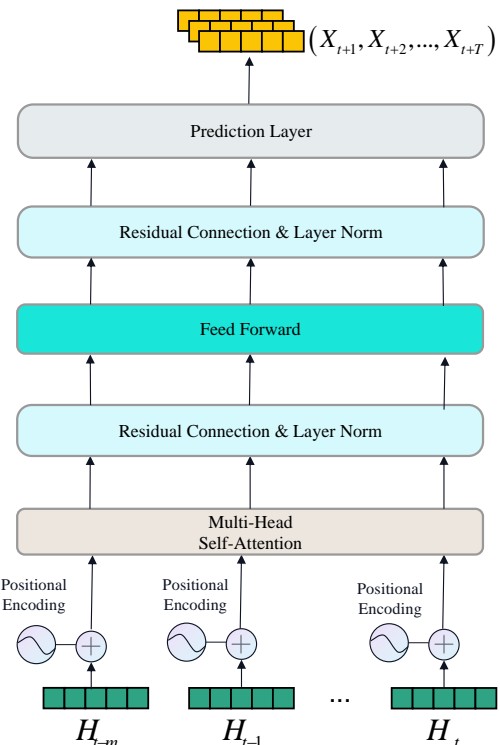

**Figure 5.** The transformer layer capturing global temporal dependence.

The temporal information is captured from different subspaces using the multi-head attention mechanism, and the ultimate output is a linear aggregation of each subspace. Define $H_h^p$ as the spatial-temporal feature matrix containing global temporal dependencies. $H_h^p$ is obtained by concatenating the feature matrix of all heads, as shown in Equation (17):

$$H_h^p = MultiHead(Q, K, V) = \|_{head=1}^{Head} \left(Attention(Q, K, V)\right) \tag{17}$$

### 4.3.3. Normalization Layer and Feed Forward Network Layer

Layer normalization is further adopted to stabilize the training of the neural network, and the residual connection is used to interact the characteristics of the lower layer with the higher layer neural network [39]. To improve the nonlinear modeling ability of the model, the features after interaction are taken as the input into the feed forward layer. Define $X_h^p$

and $\widetilde{X}_h^p$ as the outputs of the layer normalization and feed forward layer, respectively. $X_h^p$ and $\widetilde{X}_h^p$ are obtained by Equations (18) and (19):

$$X_h^p = LayerNorm\left(H_h^p + H^p\right) \tag{18}$$

$$\widetilde{X}_h^p = W_2 ReLU\left(W_1 X_h^p + b_1\right) + b_2 \tag{19}$$

where *LayerNorm* is the layer normalization operation. $W_1, W_2$ are the learnable matrices, $b_1, b_2$ are the biases.

The structure of the transformer layer is shown in Figure 5. To improve the learning capability of the model, there are two normalization sublayers and one feed forward neural sublayer in this layer.

A fully connected layer is employed as the output layer to generate the final prediction result. Define $\widetilde{Y}_h^p$ as the final output of the model, and it can be calculated by Equation (20):

$$\widetilde{Y}_h^p = W_y \widetilde{X}_h^p + b_y \tag{20}$$

where $W_y$, $b_y$ are the parameter matrix and the bias, respectively. The transformer layer focuses on different positional locations in the time series in parallel through the attention mechanism so that the global temporal dependency can be captured.

The aim of the model is to minimize the traffic prediction error. The loss function $L$ is defined as Equation (21):

$$L = \frac{1}{2}\sum\left(y - \widetilde{y}\right)^2 + \lambda\|W\| \tag{21}$$

where $y$ and $\widetilde{y}$ represent truth traffic flow and predicted value, respectively. $\lambda$ represents the vector of penalty terms. $W$ is the weight parameter matrix.

## 5. Experiments

### 5.1. Datasets

The four real-world datasets Los-loop, SZ-taxi, METR-LA, and PEMS-BAY are used in this paper to assess the prediction performance of the LGSTGCN model. Since these four datasets have been widely used in traffic flow prediction, the fairness of the data can be guaranteed. All data are detected by real road sensors to ensure that the data are representative of and characterized by the speed of vehicles on the traffic roads. The feature matrices and adjacency matrices are also included in the datasets. The brief descriptions of the four datasets are as follows:

*Los-loop.* This dataset comprises highway traffic flow from Los Angeles County. The temporal scope spans from 1 March to 7 March 2012 with traffic flow recorded at 5 min intervals. There are 207 road sensors in total. The adjacency matrix is obtained based on the distance between the road sensors.

*SZ-taxi.* This dataset was collected from Luohu District, Shenzhen, encompassing information from 156 major roads. The traffic vehicle speed is recorded at a 15 min time interval, covering the month from 1 January to 31 January 2015.

*METR-LA.* This dataset includes traffic information collected at five-minute intervals from 207 loop detectors on Los Angeles highways. The time spans of data are from 1 March to 30 June 2012.

*PEMS-BAY.* This dataset is sourced from the California Transportation Agencies Performance Measurement System and comprises traffic information collected at five-minute intervals with 325 sensors in the Bay Area. The data used for the experiments are from 1 January 2017 to 31 May 2017.

The details of the four datasets are listed in Table 1.

**Table 1.** Traffic datasets.

| Datasets | Detectors | Steps | Time Range |
|----------|-----------|-------|------------|
| Los-loop | 207 | 2016 | 3/1/2012–3/7/2012 |
| SZ-taxi | 156 | 2976 | 1/1/2015–1/31/2015 |
| METR-LA | 207 | 34,272 | 3/1/2012–6/30/2012 |
| PEMS-BAY | 325 | 52,116 | 1/1/2017–5/31/2017 |

*5.2. Baseline Methods and Evaluation Metrics*

For the datasets Los-loop and SZ-taxi, we select traditional methods (HA [32] and ARIMA [47]), a machine learning approach (SVR [48]), and deep learning approaches (GRU [27], T-GCN [26], A3T-GCN [42], DCRNN [25]) as the baseline methods.

Several models based on deep learning, including STGCN [19], SLCNN [49], DCRNN [25], Graph WaveNet [50], MRA-BGCN [51], GMAN [52], STGRAT [53], FC-GAGA [54], TSE-SC [55], STGNN [30], and STFGNN [56], are tested on the datasets METR-LA and PEMS-BAY to further verify the performance of the LGSTGCN model.

Six metrics, including Root Mean Square Error (*RMSE*), Mean Absolute Error (*MAE*), Accuracy (*Acc*), Coefficient of Determination ($R^2$), Explained Variance Score (*Var*), and Mean Absolute Percentage Error (*MAPE*) are used to assess the prediction performance of each method. The definitions of these metrics are provided below in Equations (22)–(27):

$$RMSE(y, \widetilde{y}) = \sqrt{\frac{1}{N_{pv}} \sum_{i=1}^{N_{pv}} (y_i - \widetilde{y}_i)} \tag{22}$$

$$MAE(y, \widetilde{y}) = \frac{1}{N_{pv}} \sum_{i=1}^{N_{pv}} |y_i - \widetilde{y}_i| \tag{23}$$

$$Acc(y, \widetilde{y}) = 1 - \frac{\|y - \widetilde{y}\|_F}{\|y\|_F} \tag{24}$$

$$R^2(y, \widetilde{y}) = 1 - \frac{\sum_{i=1}(y_i - \widetilde{y}_i)^2}{\sum_{i=1}(y_i - \overline{y})^2} \tag{25}$$

$$Var(y, \widetilde{y}) = 1 - \frac{Var\{y - \widetilde{y}\}}{Var\{y\}} \tag{26}$$

$$MAPE(y, \widetilde{y}) = \frac{1}{N_{pv}} \sum_{i=1}^{N_{pv}} \left| \frac{y_i - \widetilde{y}_i}{y_i} \right| \times 100\% \tag{27}$$

where $N_{pv}$ is the number of predicted value. For *RMSE*, *MAE*, and *MAPE* that evaluate the prediction errors of the model, smaller values indicate better performance. *Acc* and $R^2$ represent precision and the fitting degree between the predicted value and truth value, respectively. *Var* is used to measure the extent to which the model interprets fluctuations in the dataset. In these three metrics, higher values mean better forecasting performance.

*5.3. Experimental Settings*

In the experiments, the four datasets are partitioned into training set and test set with the ratio of 0.8:0.2. The LGSTGCN model is implemented in the Pytorch framework with an NVIDIA 1050 GPU. The task is to predict traffic speeds for the next 15, 30, and 60 min. The experimental settings of the LGSTGCN model are outlined as follows:

The training epoch is 3000, the learning rate is set to 0.001, and the number of attention heads is 4. For the Los-loop dataset, the batch size is set to 64, the number of hidden units is set to 64, and the feature size of the transformer layer is 32. For the SZ-taxi dataset, the above three hyperparameters are set to 64, 96, and 96, respectively. The METR-LA and PEMS-BAY datasets share the same three hyperparameters: 16 attention heads, 64 hidden

units, and a transformer layer feature size of 32. The Adam optimizer is employed for training the model, and an early stopping strategy is utilized.

### 5.4. Hyperparameters Analysis

Three key hyperparameters of the LGSTGCN model, including the number of hidden units, the feature size $d_{model}$ of the transformer layer, and the number of attention heads *Head*, are tested on the Los-loop dataset and SZ-taxi dataset in order to analyze their impact on the model prediction accuracy and obtain the optimal parameter settings. Figures 6 and 7 show the prediction performance with different key hyperparameter settings on the Los-loop dataset and SZ-taxi dataset. The horizontal axes represent the values of three hyperparameters and the vertical axes represent the values metrics.

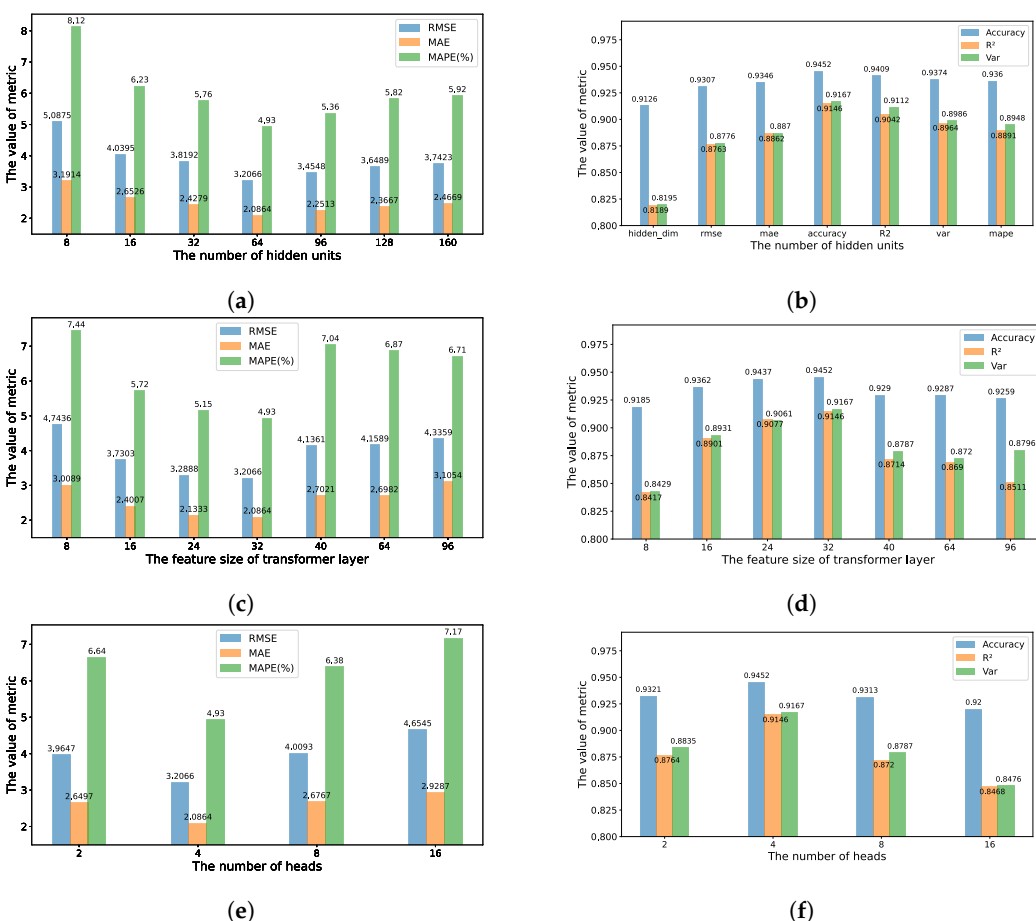

(**a**)                                      (**b**)

(**c**)                                      (**d**)

(**e**)                                      (**f**)

**Figure 6.** Evaluation metric values with different hyperparameter settings on the Los-loop dataset. (**a**) Impacts of different numbers of hidden units on *RMSE*, *MAE*, and *MAPE*. (**b**) Effects of different numbers of hidden units on Accuracy, $R^2$, and *Var*. (**c**) Influence of different feature sizes of the transformer layer on *RMSE*, *MAE*, and *MAPE*. (**d**) Influence of different feature sizes of the transformer layer on Accuracy, $R^2$, and *Var*. (**e**) Influence of different numbers of heads on *RMSE*, *MAE*, and *MAPE*. (**f**) Influence of different numbers of heads on Accuracy, $R^2$, and *Var*.

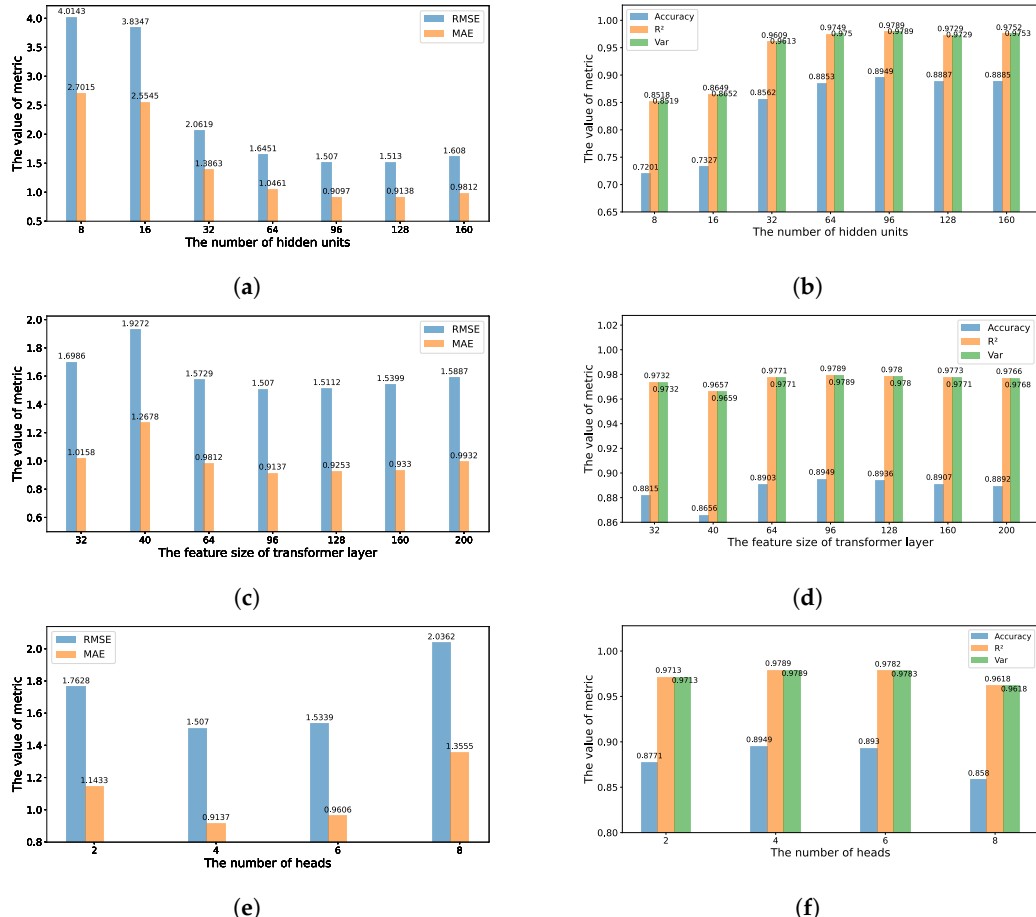

**Figure 7.** Evaluation metric values with different hyperparameter settings on the SZ-taxi dataset. (**a**) Impacts of different numbers of hidden units on *RMSE*, *MAE*, and *MAPE*. (**b**) Effects of different numbers of hidden units on Accuracy, $R^2$, and *Var*. (**c**) Influence of different feature sizes of the transformer layer on *RMSE*, *MAE*, and *MAPE*. (**d**) Influence of different feature sizes of the transformer layer on Accuracy, $R^2$, and *Var*. (**e**) Influence of different numbers of heads on *RMSE*, *MAE*, and *MAPE*. (**f**) Influence of different numbers of heads on Accuracy, $R^2$, and *Var*.

- **The number of hidden units.** The number of hidden units in a GRU model is a crucial hyperparameter influencing the model's susceptibility to overfitting. An analysis was conducted by varying the number of hidden units within the range of [8, 16, 32, 64, 96, 128, 160] to examine the impact on the model's prediction performance. Figures 6a,b and 7a,b show the variation of evaluation metrics with different numbers of hidden units for the Los-loop and SZ-taxi datasets, respectively. It shows that the model prediction performance achieves its best when the number of hidden units in the two datasets is 64 or 96.

- **The feature size $d_{model}$ of the transformer layer.** The feature size $d_{model}$ of the transformer layer represents the feature dimensional information contained in the transformer layer. $d_{model}$ is set to the values in the range of [8, 16, 24, 32, 40, 64, 96] for the Los-loop dataset and [32, 40, 64, 96, 128, 160, 200] for the SZ-taxi dataset, respectively, to analyze the effects of transformer layers containing different numbers of feature information on prediction performance. Figures 6c,d and 7c,d show the variation of metrics with different feature sizes of the transformer layer on the two datasets, respectively. It shows that the best prediction performance can be obtained with a feature size of the transformer layer of 32 on the Los-loop dataset, and 96 on the SZ-taxi dataset.

- **The number of heads.** The *Head* is also the number of subspaces in the multi-headed attention. To analyze the effects of different numbers of heads on the model's per-

formance, it is set to 2, 4, 8, and 16. Figures 6e,f and 7e,f show the values of metrics with different numbers of heads on the two datasets. It shows that the best number of attention heads is four.

The forecasting capabilities of the LGSTGCN model are improved with the growing values of key hyperparameters. However, too high values of the hyperparameters may lead to overfitting of the model. The appropriate hyperparameter settings are crucial to prediction performance, and the optimal hyperparameter values should be tested and selected for different datasets.

### 5.5. Experimental Results

Tables 2 and 3 demonstrate the forecasting results of the LGSTGCN model and the baseline models at 15, 30, and 60 min on the Los-loop and SZ-taxi datasets. HA and ARIMA methods predict traffic flows based on the assumption of smooth series, which limits their nonlinear modeling ability for traffic data. The machine learning method SVR uses a linear kernel function to automatically learn statistical laws, which provides better prediction performance compared to the traditional methods. For 15 min prediction, the RMSEs of LGSTGCN are 52.14% and 63.81% lower than those of SVR on the Los-loop and SZ-taxi datasets, while the MAE of LGSTGCN are 40.98% and 66.23% lower. LGSTGCN can achieve the best performance because of the excellent generalizability and feature extraction capability of the neural network. Deep learning approaches show better prediction performance. However, comparing with GRU, LGSTGCN has lower RMSE and higher accuracy in 15-min prediction for both datasets since LGSTGCN takes into account not only temporal dependence but also spatial dependence of traffic roads. The GNN-based deep learning models consider the spatial dependence of traffic roads and in turn achieve better prediction performance. For example, T-GCN and DCRNN combine GNN with RNN to extract the local spatial-temporal dependence. However, the LGSTGCN model shows better performance than T-GCN and DCRNN, owing to its capacity of capturing both global and local spatial-temporal dependency. Take the 30 min prediction results in Tables 2 and 3 as an example, the RMSE and MAE of LGSTGCN have decreased by 31.36%, 50.92% on the Los-loop dataset, and 30.90%, 53.73% on the SZ-taxi dataset, comparing with T-CCN. As for DCRNN, the RMSE and MAE of LGSTGCN have reduced by 34.12% and 51.23%, 19.54% and 50.81% on the two datasets, respectively. Although the A3T-GCN model uses the attention mechanism to adjust the importance of different time steps to capture global time dependency, the attention mechanism treats all time points equally and cannot identify the sequential characteristics of traffic flow data. As shown in Tables 2 and 3, comparing with A3T-GCN, the RMSE and MAE of LGSTGCN in 60 min prediction on the two datasets have reduced by 24.25%, 21.96% and 6.89%, 5.51%, while the accuracy has improved by 3.41% and 1.83%, respectively. The experimental results of LGSTGCN indicate that the spatial-temporal forecasting capabilities of the model can be enhanced by capturing global spatial-temporal features.

**Table 2.** The prediction performance of different models on the Los-loop dataset.

| Time | Models | Los-Loop | | | | | |
|------|--------|------|-----|-----|-----|-----|------|
| | | **RMSE** | **MAE** | **Acc** | $R^2$ | **Var** | **MAPE** |
| | HA | 7.3067 | 3.8782 | 0.8756 | 0.7225 | 0.7225 | 10.40% |
| | ARIMA | 10.0780 | 7.7013 | 0.8272 | * | * | 21.24% |
| | SVR | 6.6993 | 3.5352 | 0.8860 | 0.7667 | 0.7736 | 10.93% |
| 15 min | GRU | 5.1264 | 3.0194 | 0.9116 | 0.8208 | 0.8219 | 8.16% |
| | T-GCN | 5.1062 | 3.2169 | 0.9122 | 0.8229 | 0.8242 | 8.47% |
| | A3T-GCN | 5.0254 | 3.2110 | 0.9126 | 0.8232 | 0.8256 | 8.43% |
| | DCRNN | 5.1023 | 2.8447 | 0.9135 | 0.8596 | 0.8609 | 7.28% |
| | LGSTGCN | **3.2066** | **2.0864** | **0.9452** | **0.9146** | **0.9167** | **4.93%** |

**Table 2.** *Cont.*

| Time | Models | Los-Loop | | | | | |
|---|---|---|---|---|---|---|---|
| | | **RMSE** | **MAE** | **Acc** | **$R^2$** | **Var** | **MAPE** |
| | HA | 7.3067 | 3.8782 | 0.8756 | 0.7225 | 0.7225 | 10.40% |
| | ARIMA | 10.0793 | 7.7015 | 0.8272 | * | * | 21.24% |
| | SVR | 7.4739 | 3.9188 | 0.8727 | 0.7107 | 0.7191 | 12.42% |
| 30 min | GRU | 6.3616 | 3.7495 | 0.8891 | 0.7038 | 0.7107 | 10.41% |
| | T-GCN | 5.9534 | 3.8057 | 0.8961 | 0.7333 | 0.7350 | 10.58% |
| | A3T-GCN | 5.9472 | 3.6894 | 0.8974 | 0.7693 | 0.7713 | 10.26% |
| | DCRNN | 6.2034 | 3.2682 | 0.8948 | 0.7930 | 0.7959 | 9.03% |
| | LGSTGCN | **4.0867** | **2.6296** | **0.9300** | **0.8737** | **0.8752** | **6.51%** |
| | HA | 7.3067 | 3.8782 | 0.8756 | 0.7225 | 0.7225 | 10.40% |
| | ARIMA | 10.0811 | 7.7031 | 0.8272 | * | * | 21.26% |
| | SVR | 8.6882 | 4.5724 | 0.8519 | 0.6117 | 0.6246 | 15.03% |
| 60 min | GRU | 7.8011 | 4.6775 | 0.8635 | 0.5940 | 0.6087 | 13.85% |
| | T-GCN | 6.9828 | 4.6541 | 0.8769 | 0.6143 | 0.6227 | 13.13% |
| | A3T-GCN | 6.9438 | 4.3689 | 0.8788 | 0.6546 | 0.6604 | 12.28% |
| | DCRNN | 7.5975 | 3.8723 | 0.8711 | 0.6903 | 0.6940 | 11.25% |
| | LGSTGCN | **5.2326** | **3.4093** | **0.9098** | **0.8171** | **0.8279** | **8.08%** |

* indicates that the value of the evaluation metric could not be calculated. The bold number indicates the optimal value for the same evaluation metric.

**Table 3.** The prediction performance of different models on the SZ-taxi dataset.

| Time | Models | SZ-Taxi | | | | | |
|---|---|---|---|---|---|---|---|
| | | **RMSE** | **MAE** | **Acc** | **$R^2$** | **Var** | **MAPE** |
| 15 min | HA | 4.2341 | 2.7797 | 0.7049 | 0.8357 | 0.8357 | * |
| | ARIMA | 6.8044 | 4.6800 | 0.3787 | * | * | * |
| | SVR | 4.1638 | 2.7060 | 0.7098 | 0.8411 | 0.8420 | * |
| | GRU | 4.1671 | 2.7757 | 0.7096 | 0.8402 | 0.8405 | * |
| | T-GCN | 4.0862 | 2.7857 | 0.7151 | 0.8464 | 0.8468 | * |
| | A3T-GCN | 4.0785 | 2.7545 | 0.7158 | 0.8469 | 0.8470 | * |
| | DCRNN | 4.2325 | 3.0878 | 0.7053 | 0.8366 | 0.8409 | * |
| | LGSTGCN | **1.5070** | **0.9137** | **0.8949** | **0.9789** | **0.9789** | * |
| 30 min | HA | 4.2341 | 2.7797 | 0.7049 | 0.8357 | 0.8357 | * |
| | ARIMA | 6.8043 | 4.6797 | 0.3787 | * | * | * |
| | SVR | 4.2097 | 2.7828 | 0.7066 | 0.8376 | 0.8393 | * |
| | GRU | 4.1198 | 2.8359 | 0.7128 | 0.8439 | 0.8451 | * |
| | T-GCN | 4.1059 | 2.7869 | 0.7137 | 0.8449 | 0.8455 | * |
| | A3T-GCN | 4.0894 | 2.7654 | 0.7158 | 0.8461 | 0.8463 | * |
| | DCRNN | 4.1324 | 2.6218 | 0.7122 | 0.8432 | 0.8432 | * |
| | LGSTGCN | **2.0152** | **1.2896** | **0.8605** | **0.9630** | **0.9631** | * |
| 60 min | HA | 4.2341 | 2.7797 | 0.7049 | 0.8357 | 0.8357 | * |
| | ARIMA | 6.7964 | 4.6757 | 0.3789 | * | * | * |
| | SVR | 4.2738 | 2.8594 | 0.7020 | 0.8326 | 0.8351 | * |
| | GRU | 4.1564 | 2.8675 | 0.7101 | 0.8408 | 0.8421 | * |
| | T-GCN | 4.1207 | 2.7973 | 0.7126 | 0.8435 | 0.8446 | * |
| | A3T-GCN | 4.1123 | 2.7180 | 0.7197 | 0.8443 | 0.8447 | * |
| | DCRNN | 4.3953 | 3.2705 | 0.6939 | 0.8242 | 0.8321 | * |
| | LGSTGCN | **3.8291** | **2.5682** | **0.7329** | **0.8649** | **0.8652** | * |

* indicates that the value of the evaluation metric could not be calculated. The bold number indicates the optimal value for the same evaluation metric.

The MAPE values of all models on the SZ-taxi dataset are missing because the dataset has many missing values of zero and some noise figures. However, comparison on other evaluation metrics can be sufficient to analyze the performances of different models.

Table 4 shows the results of the LGSTGCN model and the baseline methods on METR-LA and PEMS-BAY datasets. STGCN can capture the traffic features by stacking multiple layers of convolution, but it focuses on the features of neighboring roads and ignores the global spatial dependence. GMAN captures global spatial-temporal correlation by using a multi-head attention mechanism. However, the attention mechanism always focuses on all time nodes and cannot capture important local spatial-temporal dependence. The LGSTGCN model captures both local and global spatial-temporal dependence. Comparing with STGCN and GMAN, the RMSE of LGSTGCN for 15 min prediction on the METR-LA dataset is reduced by 42.68% and 39.96%, respectively. STFGNN can capture global information due to the introduction of dilation convolution. However, since dilation convolution has a certain dilation limit, it makes it that so the global information of two regions that are far apart may still not be fully captured. LGSTGCN is able to capture the global dependence more completely by computing the region correlation based on all nodes. Compared to STFGNN, LGSTGCN reduces the RMSE of the 30 min prediction on the PEMS-BAY dataset by about 28.8%.

**Table 4.** The prediction performance of different models on the METR-LA and PEMS-BAY datasets.

| Time | Models | METR-LA | | | PEMS-BAY | | |
|---|---|---|---|---|---|---|---|
| | | RMSE | MAE | MAPE | RMSE | MAE | MAPE |
| 15 min | STGCN | 5.74 | 2.88 | 7.62% | 2.96 | 1.36 | 2.90% |
| | SLCNN | 5.18 | 2.53 | 6.70% | 2.90 | 1.44 | 3.00% |
| | DCRNN | 5.38 | 2.77 | 7.30% | 2.95 | 1.38 | 2.90% |
| | Graph WaveNet | 5.15 | 2.69 | 6.90% | 2.74 | 1.30 | 2.73% |
| | MRA-BGCN | 5.12 | 2.67 | 6.80% | 2.72 | 1.29 | 2.90% |
| | GMAN | 5.48 | 2.77 | 7.25% | 2.82 | 1.34 | 2.81% |
| | STGRAT | 5.07 | 2.60 | 6.61% | 2.71 | 1.29 | 2.67% |
| | FC-GAGA | 5.34 | 2.75 | 7.25% | 2.86 | 1.36 | 2.87% |
| | TSE-SC | 4.73 | 2.43 | 6.57% | 2.78 | 1.22 | 2.76% |
| | STGNN | 4.99 | 2.62 | 6.55% | 2.43 | 1.17 | 2.34% |
| | STFGNN | 4.73 | 2.57 | 6.51% | 2.33 | 1.16 | 2.41% |
| | LGSTGCN | **3.29** | **1.85** | **3.87%** | **1.52** | **1.04** | **1.90%** |
| 30 min | STGCN | 7.24 | 3.47 | 9.57% | 4.27 | 1.81 | 4.17% |
| | SLCNN | 6.15 | 2.88 | 8.01% | 3.81 | 1.72 | 3.90% |
| | DCRNN | 6.45 | 3.15 | 8.80% | 3.97 | 1.74 | 3.90% |
| | Graph WaveNet | 6.22 | 3.07 | 8.37% | 3.70 | 1.63 | 3.67% |
| | MRA-BGCN | 6.17 | 3.06 | 8.30% | 3.67 | 1.61 | 3.80% |
| | GMAN | 6.34 | 3.07 | 8.35% | 3.72 | 1.62 | 3.62% |
| | STGRAT | 6.21 | 3.01 | 8.15% | 3.69 | 1.61 | 3.63% |
| | FC-GAGA | 6.30 | 3.10 | 8.57% | 3.80 | 1.68 | 3.80% |
| | TSE-SC | 5.61 | 2.79 | 7.45% | 3.61 | 1.59 | 3.43% |
| | STGNN | 5.88 | 2.98 | 7.77% | 3.27 | 1.46 | 3.09% |
| | STFGNN | 5.46 | 2.83 | 7.46% | 3.02 | **1.39** | 3.02% |
| | LGSTGCN | **4.69** | **2.55** | **7.33%** | **2.15** | 1.42 | **2.61%** |
| 60 min | STGCN | 9.40 | 4.59 | 12.70% | 5.69 | 2.49 | 5.79% |
| | SLCNN | 7.20 | 3.30 | 9.70% | 4.53 | 2.03 | 4.80% |
| | DCRNN | 7.60 | 3.60 | 10.50% | 4.74 | 2.07 | 4.90% |
| | Graph WaveNet | 7.37 | 3.53 | 10.01% | 4.52 | 1.95 | 4.63% |
| | MRA-BGCN | 7.30 | 3.49 | 10.00% | 4.46 | 1.91 | 4.60% |
| | GMAN | 7.21 | 3.40 | 9.72% | 4.32 | 1.86 | 4.31% |
| | STGRAT | 7.42 | 3.49 | 10.01% | 4.54 | 1.95 | 4.64% |
| | FC-GAGA | 7.31 | 3.51 | 10.14% | 4.52 | 1.97 | 4.67% |
| | TSE-SC | 6.68 | 3.28 | 9.08% | 4.36 | 1.77 | 4.29% |
| | STGNN | 6.94 | 3.49 | 9.69% | 4.20 | 1.83 | 4.15% |
| | STFGNN | **6.40** | **3.18** | **8.81%** | **3.74** | **1.66** | **3.77%** |
| | LGSTGCN | 7.26 | 4.31 | 10.89% | 4.43 | 2.56 | 4.98% |

The bold number indicates the optimal value for the same evaluation metric.

The forecasting capabilities of LGSTGCN has excellent performance in shorter-term predictions compared with traffic spatial-temporal prediction models in recent years (except for one MAE metric in the 30 min prediction of the PEMS-BAY dataset, which is slightly larger), such as in the 15 min prediction of the METR-LA dataset, where LGSTGCN reduces RMSE, MAE, and MAPE compared to the optimal results in the baseline by 30.44%, 28.02%, and 44.55%, and in the 30 min prediction by 14.1%, 9.89%, and 1.74%, respectively.

The forecasting capabilities of LGSTGCN are poor in the 60 min prediction on both datasets. The main reason is that the temporal and spatial correlations of road traffic flow become more complex when the prediction step is increased. Since LGSTGCN captures both local and global features in time and space, the global noise data in long-term prediction lead to larger overall error of the model. In the 15 min short-term prediction, the global noise data are less, and the comprehensive feature capturing capacity of the LGSTGCN model makes its prediction performance better, comparing with all baseline models.

### 5.6. Ablation Study

The ablation experiments of the LGSTGCN model on the two datasets are conducted to evaluate the effectiveness of key components. The five variants of the LGSTGCN model are designed as follows:

- **NP_GS:** In this variant, the graph attention residual network layer, which is used to capture global spatial dependency, is removed from LGSTGCN.
- **NO_LST:** This is a variant without T-GCN, which is used to capture real-time local spatial-temporal dependency.
- **NO_GT:** The transformer layer that can capture global temporal dependence is removed.
- **NO_PE:** The positional encoding of the transformer layer that can recognize the positional characteristics of the sequence is removed.
- **NO_NC:** This is a variant exchanging the graph attention residual network layer and T-GCN layer. Its graph convolution operation aggregates neighborhood information only according to static initial neighborhood features, ignoring the dynamic neighborhood features.

Tables 5 and 6 list the comparison of five LGSTGCN variants, and the conclusions are as follows:

- **Cross-regional spatial dependency is effective.** The LGSTGCN model has a smaller prediction error compared to the NO_GS variant on both datasets. It indicates that cross-regional spatial dependence is effective to increase the prediction performance.
- **Real-time local spatial-temporal dependency is significant.** As shown in Table 5, for the Los-loop dataset, the forecasting capability of the NO_LST is worse than that of the LGSTGCN model. It means that the lack of local spatial-temporal information can lead to degradation in prediction performance. For the SZ-taxi dataset, the LGSTGCN model has the best performance in both 15 min and 30 min prediction. However, NO_LST is better than LGSTGCN in 60 min prediction. It may be because the correlations between the traffic flow data are reduced as the prediction horizon increases on the one hand. On the other hand, the wrong information at the current time may be learned due to the continuous missing values and noise figures existing in the dataset.
- **Global temporal dependency is necessary.** The forecasting capability of the NO_GT on the two datasets degrades significantly, which shows that the global temporal dependence module in the LGSTGCN model can improve the forecasting capabilities greatly.
- **Positional information is effective.** The NO_PE without positional encoding is unable to identify the order between related sequences when processing global temporal information. Thus, it is necessary to consider sequence position information.

- **Dynamic neighborhood correlation is important.** The NO_NC only aggregates information of neighborhood nodes based on static spatial distance and cannot capture the dynamic features. Thus, the prediction results of the NO_NC are poor. It indicates that it is important to take the influence of neighborhood nodes into account.

**Table 5.** The results of ablation study on the Los-loop dataset.

| Time | Models | Los-Loop | | | | | |
|---|---|---|---|---|---|---|---|
| | | **RMSE** | **MAE** | **Acc** | $R^2$ | **Var** | **MAPE** |
| 15 min | NO_GS | 3.6489 | 2.3667 | 0.9374 | 0.8964 | 0.8986 | 5.82% |
| | NO_LST | 3.8702 | 2.2495 | 0.9335 | 0.8883 | 0.8886 | 5.42% |
| | NO_GT | 4.8423 | 3.0542 | 0.9158 | 0.8029 | 0.8046 | 7.93% |
| | NO_PE | 3.5384 | 2.3052 | 0.9394 | 0.9014 | 0.9052 | 5.44% |
| | NO_NC | 4.3329 | 2.7929 | 0.9256 | 0.8630 | 0.8637 | 6.82% |
| | LGSTGCN | **3.2066** | **2.0864** | **0.9452** | **0.9146** | **0.9167** | **4.93%** |
| 30 min | NO_GS | 4.4182 | 2.8233 | 0.9241 | 0.8605 | 0.8628 | 7.49% |
| | NO_LST | 4.8201 | 2.8582 | 0.9171 | 0.8405 | 0.8409 | 7.25% |
| | NO_GT | 5.8756 | 3.6424 | 0.8988 | 0.7768 | 0.7831 | 9.96% |
| | NO_PE | 4.5248 | 2.8730 | 0.9222 | 0.8530 | 0.8540 | 7.06% |
| | NO_NC | 4.6841 | 2.9214 | 0.9194 | 0.8442 | 0.8454 | 7.32% |
| | LGSTGCN | **4.0867** | **2.6296** | **0.9300** | **0.8737** | **0.8752** | **6.51%** |
| 60 min | NO_GS | 6.2143 | 4.1995 | 0.8929 | 0.7483 | 0.7680 | 10.26% |
| | NO_LST | 5.9255 | 3.7061 | 0.8978 | 0.7737 | 0.7765 | 9.49% |
| | NO_GT | 6.7056 | 4.2913 | 0.8827 | 0.6787 | 0.6837 | 12.32% |
| | NO_PE | 5.6259 | 3.6492 | 0.9031 | 0.7812 | 0.7858 | 9.49% |
| | NO_NC | 5.4947 | 3.6249 | 0.9055 | 0.7914 | 0.8040 | 8.74% |
| | LGSTGCN | **5.2326** | **3.4093** | **0.9098** | **0.8171** | **0.8279** | **8.08%** |

The bold number indicates the optimal value for the same evaluation metric.

**Table 6.** The results of ablation study on the SZ-taxi dataset.

| Time | Models | SZ-Taxi | | | | | |
|---|---|---|---|---|---|---|---|
| | | **RMSE** | **MAE** | **Acc** | $R^2$ | **Var** | **MAPE** |
| 15 min | NO_GS | 2.1439 | 1.4201 | 0.8505 | 0.9575 | 0.9575 | * |
| | NO_LST | 2.6827 | 1.6417 | 0.8130 | 0.9339 | 0.9340 | * |
| | NO_GT | 4.0549 | 2.7526 | 0.7173 | 0.8488 | 0.8490 | * |
| | NO_PE | 1.6548 | 1.0631 | 0.8846 | 0.9746 | 0.9746 | * |
| | NO_NC | 2.9412 | 1.9896 | 0.7949 | 0.9204 | 0.9205 | * |
| | LGSTGCN | **1.5070** | **0.9137** | **0.8949** | **0.9789** | **0.9789** | * |
| 30 min | NO_GS | 2.3694 | 1.5592 | 0.8348 | 0.9482 | 0.9482 | * |
| | NO_LST | 2.9076 | 1.7041 | 0.7973 | 0.9224 | 0.9224 | * |
| | NO_GT | 4.1167 | 2.7932 | 0.7130 | 0.8440 | 0.8443 | * |
| | NO_PE | 3.5735 | 2.3970 | 0.7513 | 0.8829 | 0.8835 | * |
| | NO_NC | 3.7193 | 2.4928 | 0.7406 | 0.8727 | 0.8728 | * |
| | LGSTGCN | **2.0152** | **1.2896** | **0.8605** | **0.9630** | **0.9631** | * |
| 60 min | NO_GS | 3.6902 | 2.4640 | 0.7426 | 0.8745 | 0.8746 | * |
| | NO_LST | **3.6877** | **2.3506** | **0.7428** | **0.8749** | **0.8750** | * |
| | NO_GT | 4.1930 | 2.9217 | 0.7075 | 0.8382 | 0.8396 | * |
| | NO_PE | 3.8855 | 2.5758 | 0.7290 | 0.8612 | 0.8614 | * |
| | NO_NC | 3.9767 | 2.6809 | 0.7226 | 0.8545 | 0.8546 | * |
| | LGSTGCN | 3.8291 | 2.5682 | 0.7329 | 0.8649 | 0.8652 | * |

* indicates that the value of the evaluation metric could not be calculated. The bold number indicates the optimal value for the same evaluation metric.

### 5.7. Analysis of Visualization

To better elucidate the LGSTGCN model, we specifically choose two roads from each dataset and visualize the prediction outcomes across various horizons. We utilize the

full-day data on 7 March 2021 from the Los-loop dataset and traffic flow data spanning 28 January 2015 to 31 January 2015 from the SZ-taxi dataset. The visualizations at 15 min, 30 min, and 60 min intervals are presented in Figure 8. The LGSTGCN model demonstrates impressive predictive capabilities for nonlinear traffic speed across different time intervals. Notably, the prediction results exhibit non-smooth patterns in the 15 min prediction, indicating the model's accuracy in capturing short-term traffic variations. As the prediction horizon extends, the results gradually smooth out while still maintaining a consistent trend with actual traffic flow. Furthermore, the visualization results on the Los-loop dataset demonstrate that the model is adaptive to the situations involving sudden drops in traffic speed, making it valuable for predicting traffic congestion.

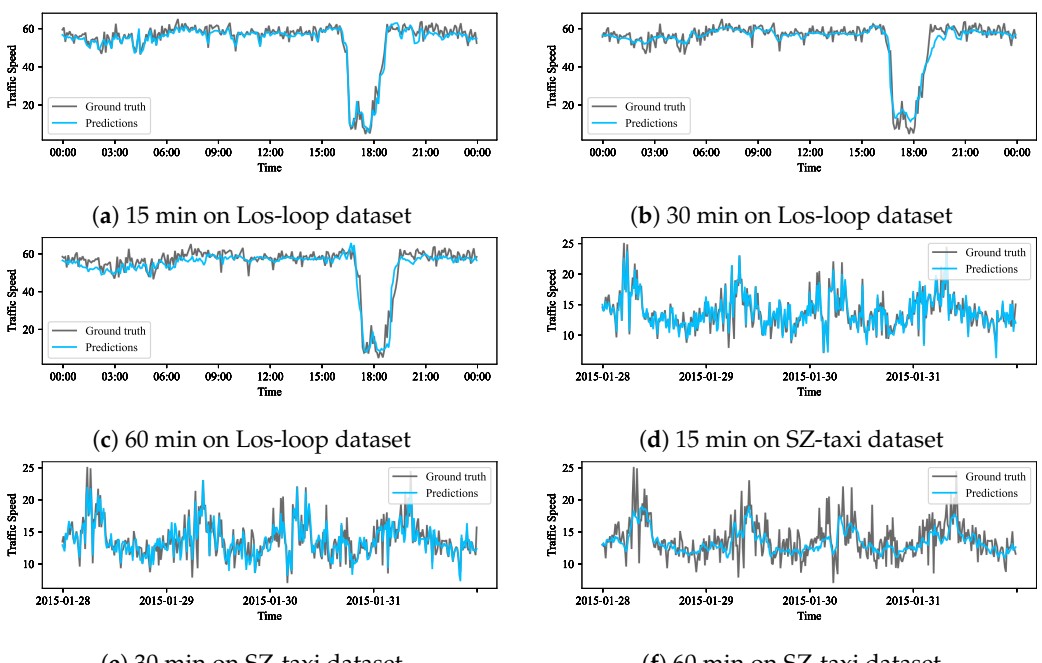

**Figure 8.** Visualization results with different prediction horizons on the Los-loop and SZ-taxi datasets.

## 6. Conclusions

A novel framework called Local-Global Spatial-Temporal Graph Convolutional Network (LGSTGCN) for traffic prediction is presented in this paper. In the model, a graph attention residual network layer is proposed to capture global spatial dependency. Moreover, a T-GCN module is used to extract local spatial-temporal dependency adaptively. In addition, a transformer layer is introduced to capture the global temporal features of all traffic road nodes. Therefore, the LGSTGCN model can extract local and global traffic flow information in both temporal and spatial terms. The experimental results on all datasets show that the LGSTGCN model outperforms the existing traffic prediction methods. It cannot only adapt to short-term changes in traffic speed but also capture long-term temporal dependence. Furthermore, the LGSTGCN model can maintain the cross-regional global spatial dependence and capture the real-time local spatial information. The analysis on the ablation study shows that both local and global spatial-temporal dependence is necessary for traffic prediction. The results of hyperparameter experiments show that different hyperparameter settings are required for different datasets to achieve optimal performance.

The future research mainly focuses on the traffic prediction modeling involving external information, such as weather, holidays, etc. These external factors allow the model to learn more realistic and richer traffic features. Research on prediction methods that further improves prediction performance is also an important challenging issue.

**Author Contributions:** Conceptualization, X.Z. and Z.C.; methodology, X.Z. and Z.C.; software, F.Y.; validation, X.Z., Z.C. and F.Y.; formal analysis, X.Z.; investigation, Z.C. and F.Y.; resources, Z.C.; data curation, F.Y.; writing—original draft preparation, X.Z. and Z.C.; writing—review and editing, X.Z. and S.W.; visualization, Z.C.; supervision, X.Z. and S.W.; project administration, X.Z. and S.W.; funding acquisition, X.Z. All authors have read and agreed to the published version of the manuscript.

**Funding:** This research was supported in part by the National Natural Science Foundation of China under Grant (62376089 and 62202147) and in part by the Hubei Provincial Science and Technology Plan Project under Grant (2023BCB04100).

**Data Availability Statement:** Los-loop, SZ-taxi, METR-LA, and PEMS-BAY datasets can be obtained at https://github.com/lrhan321/LGSTGCN-2023.git (accessed on 29 December 2023).

**Conflicts of Interest:** The authors declare no conflicts of interest.

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
