# Peer review of "Local-Global Spatial-Temporal Graph Convolutional Network for Traffic Flow Forecasting"

_electronics, doi:10.3390/electronics13030636_

Round 1
Reviewer 1 Report
Comments and Suggestions for Authors
1. The introductory part of the article was very well grounded in theory by the authors. The article presents an essential novel element: a novel framework for traffic prediction based on a Local-Global Spatial-Temporal Graph Convolutional Network (LGSTGCN). The contribution part is very well highlighted, as well as the description of the organization of the article from the introduction section.
2. Based on the similarity coefficients report in the attachment, please make the changes concerning sources 1 (https://www.mdpi.com/2076-3417/10/4/1509) and 2 (https://www.nature.com/articles/s41598-023-48579-3).
3. The conclusions are requested to be more systematized concerning the title of the paper and the experimental results obtained.
4. In the end, congratulations to the authors for all their hard work producing this scientific material.

Comments on the Quality of English LanguageModerate editing of the English language is needed.
Reviewer 2 Report
Comments and Suggestions for Authors
This paper focuses on capturing both global spatial-temporal dependencies and local spatial-temporal dependency to enhance the accuracy of traffic flow prediction. The authors propose a novel deep learning model, LGSTGCN, comprising distinct layers tailored to effectively capture global and local spatial-temporal dependencies. Through rigorous evaluation using four diverse datasets, the authors demonstrate that, in the majority of cases, their model surpasses existing traffic prediction methods.
To enhance the overall quality of the paper, it is suggested that the authors articulate the significance of cross-regional spatial dependence more explicitly, possibly through the incorporation of additional illustrative examples. Furthermore, a thorough analysis of why the prediction performance of LGSTGCN is suboptimal in the 60-minute prediction scenario would be valuable. Understanding these limitations is crucial for users knowing both the strengths and weaknesses of LGSTGCN.
Comments on the Quality of English Language
Additionally, the authors are advised to conduct a proofreading of their paper to rectify any grammatical errors and eliminate any instances of repeated sentences.
Reviewer 3 Report
Comments and Suggestions for Authors
This paper demonstrated a traffic forecasting method with Local-Global Spatial-Temporal Graph Convolutional Network method. The author utilizes a transformer layer to capture long-term parameters to adapt the use case in complicated traffic flow predictions. This is a realistic application in our daily life. More detailed comments are listed below:
1. The transformer layer has a very similar effect to LSTM, what are their difference and how do you compare their performance. From Line 64-70, I don’t get the answer.
2. In Figure1b, the plot has a couple curves, which doesn’t have any legend to show what do the corresponding to, the context is not explaining them neither. Please make the figure more informative.
3. The trickiest section of this paper located on its transformer layer, while I didn’t find a detailed description about how the layer works in Methodology section.
4. There are some related work are working on traffic predictions, can you compare your result with others? For example, what is the prediction precision from others works for the next 30 and 60 minutes.
5. When you are choosing the 4 databases, what is their specialty, and how do they stand for most of the realistic cases?
Comments on the Quality of English Language
It is fine for me to understand, I think it is ok.
Reviewer 4 Report
Comments and Suggestions for Authors
This manuscript presents a comprehensive exploration of the LGSTGCN model for time-series prediction in complex systems. The authors meticulously detail the model's architecture, emphasizing its unique aspects in handling spatiotemporal data. Overall, this work stands as a promising contribution and would elevate its significance and impact within the research community. Here are some remarks to consider:
- Please expand the discussion on the dataset selection criteria and how it represents real-world scenarios, including any potential biases or limitations.
- Please consider a more in-depth comparison of the LGSTGCN model with other contemporary approaches, emphasizing its unique advantages.
- Please consider discussing the impact of external factors, such as environmental changes or socio-economic events, on the model's predictive capabilities.
Round 2
Reviewer 3 Report
Comments and Suggestions for Authors
This version has addressed all my concern commented earilier.